# Constructing Micro Knowledge Graphs from Technical Support Documents

Atul Kumar, Nisha Gupta, and Saswati Dana

IBM Research - India
G2 Block, Manyata Embassy, Outer Ring Rd, Nagavara, Bengaluru, India 560045
{kumar.atul, nisgup97, sdana027}@in.ibm.com

**Abstract.** Short technical support pages such as IBM Technotes are quite common in technical support domain. These pages can be very useful as the knowledge sources for technical support applications such as chatbots, search engines and question-answering (QA) systems. Information extracted from documents to drive technical support applications is often stored in the form of Knowledge Graph (KG). Building KGs from a large corpus of documents poses a challenge of granularity because a large number of entities and actions are present in each page. The KG becomes virtually unusable if all entities and actions from these pages are stored in the KG. Therefore, only key entities and actions from each page are extracted and stored in the KG. This approach however leads to loss of knowledge represented by entities and actions left out of the KG as they are no longer available to graph search and reasoning functions. We propose a set of techniques to create micro knowledge graph (micrograph) for each of such web pages. The micrograph stores all the entities and actions in a page and also takes advantage of the structure of the page to represent exactly in which part of that page these entities and actions appeared, and also how they relate to each other. These micrographs can be used as additional knowledge sources by technical support applications. We define schemas for representing semi-structured and plain text knowledge present in the technical support web pages. Solutions in technical support domain include procedures made of steps. We also propose a technique to extract procedures from these webpages and the schemas to represent them in the micrographs. We also discuss how technical support applications can take advantage of the micrographs.

## 1   Introduction

Using Knowledge Graphs for storing domain specific knowledge extracted from unstructured and semi-structured documents is common in several domains including technical support  [8,12,11,6]. KG stores knowledge in structured form where entities are linked with each other by certain edges representing relationship among them  [1,4,10,7]. Knowledge graphs not only make it easy to lookup the desired information, the relations represented in form of direct edges and hierarchies also support reasoning and therefore answering complex questions. In technical support domains - both hardware and software - knowledge graphs are

used extensively [6,11,12]. When data is available in semi-unstructured and/or plain text forms, developing a knowledge graphs which provides high performance on querying is a challenging task [8]. Architectural design of schema depends on the purpose and has significant impact on query performance [5,9]. We propose to augment knowledge bases that are developed using a popular category of technical support documents. These documents are relatively short web pages providing solutions to one or more problems or contain information to perform some task such as downloading and installing a software extension. Examples of such documents are IBM Technotes for DB2 and other IBM hardware and software products (Some example technotes can be browsed at [3]). We have also worked with similar technical support web pages for other non IBM hardware and software products. These technical documents are intended to be used by human end-users as self help guides. They are referred to by technical support human agents to answer customer queries and to resolve or troubleshoot their problems. Each of these documents contains rich information such as problem description with symptoms, diagnostic steps, solutions, and also the applicable constraints such as relevant hardware platforms and operating systems. Complexity of the information and it's hierarchy depends on the type of technical document, issue, solution steps and other details present in the page. Other than the usual challenges such as entity extraction and linking, it is also important to design a schema that can represent all the entities and their relations in a domain in a reasonable manner and can efficiently support knowledge graph tasks such as population, search, lookup, deletion etc. Every technical support page contains several dozen entities if not hundreds. Not all entities are equally important in the context of the primary topic of that page. Therefore, a typical choice of schema for building a knowledge graph from such pages is to identify key entities and actions/symptoms from these pages and store the URLs/identities of the pages as the solution nodes in the graph. A technical support application can present the URL(s) of the relevant page(s) containing solution/answer. User can then read and use the information present in the page without any additional help as these pages are short. This approach serves reasonably well for question-answer (QA) and search applications. But if there are multiple similar looking pages providing the solution for a problem related to similar entities (but differing on one or more situations) or if the page itself contains different solutions based on some condition (eg, OS version or hardware platform), then it is desirable to return most relevant page (or part of the page) to the user for a query. Understanding the solution presented in these pages at a fine grained level including various constraints, conditions and steps involved in the solutions (if any) is necessary for not only presenting the most specific page/part but also to support applications that require reasoning. For example, a chatbot that can ask a followup question to user in order to disambiguate between several possible answers. Or, an application that provides step by step guidance including conditional steps. Identifying individual steps from a solution procedure present in a support page is also useful for automating the support function where steps can be executed on user's behalf by the system. In the next section, we present

algorithms and techniques to extract knowledge (including entities, actions and procedures) from short technical support web pages to construct the individual micrographs for these pages. We also show an example micrograph for a real technical support document (an IBM DB2 Technote web page).

## 2 Micrograph Construction

Technical support web pages use different structures across products and manufacturers/vendors. But a closer examination reveals many similarities among these pages. For example, they all have a title that contains important entities and actions. There are fixed number of document types such as troubleshooting, FAQ, Howto etc. These pages have sections that include symptoms, diagnostic steps, solution, constraints, links to related information and references. Before using the automatic micrograph generation system to process a document corpus, a manual step of examining a representative subset of that corpus is required. We create some meta-information in a predefined format for the given document corpus. The meta-information contains items such as the number and names of different document types present in the corpus, section headings expected in each type and their mappings to the generic types found in technical support documents, list of entities that are used to denote constraints (eg, operating system, hardware platform), and dictionaries of product specific entities and action verbs. We have defined a generic set of schema to represent commonly seen types of documents in the technical support domain. If we come across a new document type in a document corpus then we create a new schema to represent that type. The high level architecture of the micrograph construction system is shown in Fig. 1.

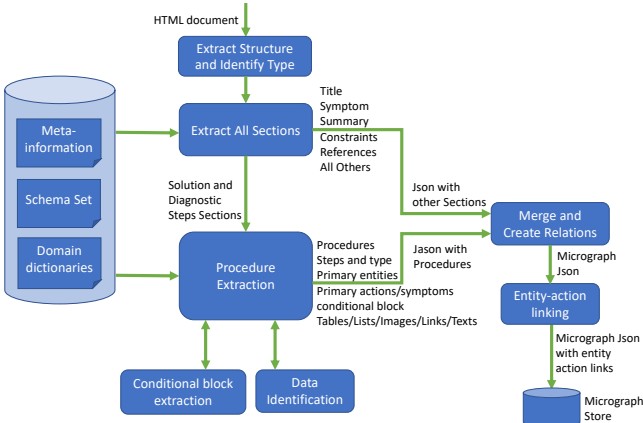

**Fig. 1.** Architecture of the Micrograph Construction System

We use the meta-information and HTML structure to identify sections in a page. All plain-text, structured elements and non text elements in a section are extracted for all sections. Contents of solution and diagnostic step sections, if any, are passed to the procedure extraction module that is described in details in subsection 2.1. Output of the procedure extraction modudle is then merged and linked with the contents of other sections. We use a custom entity extraction and linking algorithm to extract and link entities and actions from all text elements. This step is optional. Finally the constructed micrograph json is ingested in a graph store.

An example micrograph for a DB2 Technote page [2] is shown in Fig. 2

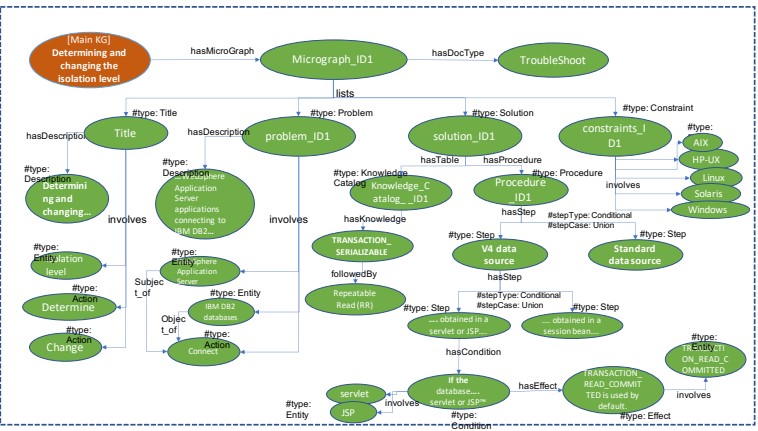

**Fig. 2.** An example micrograph constructed from a DB2 technote page

## 2.1   Procedure Extraction

This module takes as input the contents of Solution and Diagnostic Step sections in HTML format. Output of this module is the array of all procedures as a json. A procedure consists of a sequence of steps where each step is described by: (i) Type of step (i.e. Sequential or conditional), (ii)Conditional block if any, (iii) Nested steps/procedures, and (iv) Step's content.

At a high level, procedure extraction algorithm is described as follows:

– Find the different sets of steps within the solution section using information such as HTML Tags, patterns across the contents of the tags, relationship between the tags and documents and document meta-information along with predefined set of word vectors.
– Find the first set of steps among all sets such that the steps from this set covers the entire solution section as the parent stepsets. This is done based on the position of the steps in the document.

- Extract each step of the parent stepsets from the document along with its title and content using HTML navigation.
- Process the title of the steps using entity extraction and linking to find the type of the steps. Steps could be sequential or conditional.
- Repeat the above methods to similarly find the nested steps and step type within each step content. This recursive approach extracts every possible nested steps to the finest granular level of the solution section.
- Identify the type of the textual/pictorial content of the steps.
- Extract the conditional block from the textual content of the steps. Every conditional block has a conditional statement along with the effect statement of the condition. Identify such blocks using PoS tag and DEP tag from the parse tree of the textual content.

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
