# OpenReview forum: "Constructing Micro Knowledge Graphs from Technical Support Documents"
_eswc-conferences.org/ESWC/2021/Conference/Industry_Track — ESWC 2021 Industry_

### Official Review · ~Aneta_Koleva1 · 2021-04-16
**KG representation of steps and actions from technical support documents**

**Rating:** 6
**Confidence:** 3

**Review:**

This paper presents an idea for representing steps from technical support documents as a knowledge graph. The motivation for such representation is to make the search for a specific step easier and more accessible to a chatbot or search engines.
Overall, the paper is well written, however there are some grammatical errors which should be corrected. The authors mention that KGs have been used for representing documents from the technical support domain, so I assume that the novelty is in that they are proposing extraction of KGs from specific type of technical support documents (IBM Technotes).  Details about the techniques used for creating such KG are missing, there is one figure with an example of a KG but no explanation for how it was constructed, the concrete steps, what are the different types of entites?

**Pros** - interesting idea, well motivated

**Cons** - lack of details about the techniques applied for creating the KG, e.g., how are key entities and actions choosen?

---

### Official Review · ~Maria_Husmann1 · 2021-04-21
**Good use case, maturity unclear**

**Rating:** 7
**Confidence:** 3

**Review:**

The paper describes the use of knowledge graphs for technical support documents. The use case is explained very well and is easy to understand.

The paper mentions techniques to create micro knowledge graphs for support web pages. No information is given on any evaluation or usage within the company. So the maturity of the work is somewhat unclear. Were the techniques used on a bigger data set within the company? Is there any quantitative data? What were the lessons learned? Or is this work still at concept stage?

---

### Official Review · ~Victor_Charpenay1 · 2021-04-23
**The abstract has little detail but authors should get the chance to answer questions at the conference**

**Rating:** 6
**Confidence:** 3

**Review:**

The presented work is a transformation from IBM's product documentation (technotes) to a KG that can be used e.g. by chatbots. I have two main remarks.

First, there is very little detail on the exact nature of the work that has been done. The technotes used as input are already fairly well structured; why can't a chatbot already query them in a document store? The authors say they "propose a set of techniques" to build the KG but no further detail is given; do these technique involve database migration, NLP, manual work...?

My second remark is about the term "micrograph":  a KG intends to integrate smaller pieces of information rather than to split information into smaller pieces. Is it necessary to isolate each technote in its own micrograph? Aren't problems and solutions partially overlapping across notes?

The abstract leaves many questions open but authors could probably answer them at the conference, hence the decision to accept.

Minor typos:
 - "it desirable": it is
 - "its desirable": it is
 - "for user's query": for the user's query